# A generalized multipath delayed-choice experiment on a large-scale quantum nanophotonic chip

Xiaojiong Chen[1,10], Yaohao Deng[1,10], Shuheng Liu[1,10], Tanumoy Pramanik [1,2], Jun Mao[1], Jueming Bao[1], Chonghao Zhai[1], Tianxiang Dai[1], Huihong Yuan[1], Jiajie Guo[1], Shao-Ming Fei[3], Marcus Huber [4,5], Bo Tang[6], Yan Yang[6✉], Zhihua Li[6], Qiongyi He[1,2,7,8,9✉], Qihuang Gong[1,2,7,8,9✉] & Jianwei Wang [1,2,7,8,9✉]

Bohr's complementarity is one central tenet of quantum physics. The paradoxical wave-particle duality of quantum matters and photons has been tested in Young's double-slit (double-path) interferometers. The object exclusively exhibits wave and particle nature, depending measurement apparatus that can be delayed chosen to rule out too-naive interpretations of quantum complementarity. All experiments to date have been implemented in the double-path framework, while it is of fundamental interest to study complementarity in multipath interferometric systems. Here, we demonstrate generalized multipath wave-particle duality in a quantum delayed-choice experiment, implemented by large-scale silicon-integrated multipath interferometers. Single-photon displays sophisticated transitions between wave and particle characters, determined by the choice of quantum-controlled generalized Hadamard operations. We characterise particle-nature by multimode which-path information and wave-nature by multipath coherence of interference, and demonstrate the generalisation of Bohr's multipath duality relation. Our work provides deep insights into multidimensional quantum physics and benchmarks controllability of integrated photonic quantum technology.

[1] State Key Laboratory for Mesoscopic Physics, School of Physics, Peking University, Beijing, China. [2] Beijing Academy of Quantum Information Sciences, Beijing, China. [3] School of Mathematical Sciences, Capital Normal University, Beijing, China. [4] Institute for Quantum Optics and Quantum Information – IQOQI Vienna, Austrian Academy of Sciences, Vienna, Austria. [5] Vienna Center for Quantum Science and Technology, Atominstitut, TU Wien, Vienna, Austria. [6] Institute of Microelectronics, Chinese Academy of Sciences, Beijing, China. [7] Frontiers Science Center for Nano-optoelectronics & Collaborative Innovation Center of Quantum Matter, Peking University, Beijing, China. [8] Collaborative Innovation Center of Extreme Optics, Shanxi University, Taiyuan, Shanxi, China. [9] Peking University Yangtze Delta Institute of Optoelectronics, Nantong, Jiangsu, China. [10] These authors contributed equally: Xiaojiong Chen, Yaohao Deng, Shuheng Liu. ✉email: yyang10@ime.ac.cn; qiongyihe@pku.edu.cn; qhgong@pku.edu.cn; jww@pku.edu.cn

Famous double-slit or double-path experiments, implemented in a Young's or Mach–Zehnder interferometer, have confirmed the dual nature of quantum matters[1]. When a stream of photons[2], neutrons[3], atoms[4], or molecules[5], passes through two narrow slits, either wave-like interference fringes build up on a screen, or particle-like which-path distribution can be ascertained. These quantum objects exhibit both wave and particle properties but exclusively, depending on the way they are measured[1]. In the equivalent Mach–Zehnder configuration, quantum objects display either wave or particle nature in the presence or absence of a beamsplitter, respectively, where beamsplitter represents the choice of measurement apparatus[6]. Wheeler further proposed a Gedanken experiment[7], in which the choice of particle or interference measurement is made after the object has already entered the interferometer, so as to exclude the possibility of predicting with which measurement it will be confronted. Delayed-choice experiments have enabled significant demonstrations of the genuine two-path duality of different quantum objects[8–15]. Moreover, a quantitative description of two-slit duality relation was initialized in Wootters and Zurek's seminal work[6] and then formalized by Greenberger, Yasin, Jaeger, and Englert[16–18] as $\mathcal{D}^2 + \mathcal{V}^2 \le 1$, where $\mathcal{D}$ is the distinguishability of which-path information (a measure of particle-property), and $\mathcal{V}$ is the contrast visibility of interference (a measure of wave-property). This double-path duality relation has been tested in pioneer experiments[19] and in delayed-choice measurements[11,13].

Since the birth of quantum mechanics, it has long been of fundamental interests to understand multipath interference of quantum mechanical wavefunction in complex quantum systems[20–25]. Quantum nature represented as the principles of complementarity and superposition however remains ambiguous in multipath interferometric quantum systems[17,26–30]. Figure 1a sketches a general multipath Mach–Zehnder interferometric delayed-choice experiment with a single photon. Whether the photons take one or multiple paths to a given detector depends on the absence or presence of the second $d$-mode beamsplitter. In a delayed-choice experiment, the photons can either take all $d$ paths simultaneously (multipath wave character), or one of the $d$ paths (multimode particle character), or everything in between, determined in a delayed manner by the choice of the $d$-mode particle measurement or interference measurement. In contrast to the double-slit implementation, the duality relation well describes the wave-particle complementarity, however, it cannot be simply generalized in the multipath experiment[26–30]. There are several major open questions remaining: Can Bohr's duality relation still hold in the multipath interferometric experiment? Are there any good measures of multipath wave and multimode particle properties that are accessible in experiment? Does single photon preserve the inherent dual nature in the multipath delayed-choice scenario? Revealing these unknowns are essential to understand multimode quantum superposition and quantization in complex quantum systems. Apart from fundamental interests, the characterization of multimode quantum properties in controllable systems may provide the ground of developing multidimensional quantum technology[31]. For example, quantifying multimode coherence from sophisticated multipath interference patterns is of practical significance[32], in the light of recent reassessment of coherence as a key resource in quantum information[33,34], while it has always been a core concept underlying the theory of quantum mechanics. In general, when single photons pass through multiple paths, a superposition of multiple modes naturally forms a *qudit* state. Promising prospects of qudit-based quantum applications have

been well acknowledged, such as noise-robust multi-dimensional entanglement[35,36], resource-efficient quantum computations, and simulations[37,38], and high-capacity quantum communications[39,40]; however, the deep understanding of the most elementary physics of multidimensional quantum systems is highly demanded. Any explorations of multi-dimensional quantum science and technology strongly rely on the quantum platform that can be operated with high-level controllability, efficiency, and versatility[31]. It is here the integrated-optics implementation provides one of the most competitive multidimensional quantum platforms[41].

Here, we report a quantum delayed-choice multipath experiment and demonstrate a generalization of wave-particle duality relation. The wave-particle nature of single photons propagating in a $d$-path ($d$ up to 8) interferometer is observed, determined in a delayed manner by the state of a $d$-mode quantum-controlled beamsplitter. Qualitative wave-particle transitions having a 0.99 fidelity of theoretical and experimental results, and quantitative multipath duality relation are both confirmed in the context of delayed-choice. We show that quantum coherence is a good measure of the wave-property in $d$-path interference, and the amount of coherence can be directly probed from interference patterns, without accessing the density matrix. The $d$-mode which-path information is identified, and quantum randomness is efficiently generated. All demonstrations are enabled by realizing a multipath delayed-choice interferometric system on a large-scale silicon nanophotonic quantum chip that monolithically integrates 355 optical components and 95 phase-shifters.

## Result
### Scheme of the generalized multipath delayed-choice experiment.
Figure 1a shows a diagram of general $d$-path Mach–Zehnder interferometer ($d$-MZI) consisting of $d$ arms (paths) and two $d$-mode beamsplitters ($d$-BSs). An array of individually reconfigurable phases of $\{\theta_k\}_{k=0}^{d-1}$ are applied on the $d$-path between the two beamsplitters of $d$-BS1 and $d$-BS2. To implement the $d$-BSs, a general scheme is to nest $(d^2 - d)/2$ standard 2-BSs (see a bulk-optic example in Fig. 1b). For simplicity, we consider the case that $d$-BSs are well balanced, thus the state emerging from the $d$-BS1 is a maximally coherent state as $\rho_0 = |\Psi\rangle_{00}\langle\Psi|$ and $|\Psi\rangle_0 = \frac{1}{\sqrt{d}}\sum_{k=0}^{d-1}|k\rangle$, where $\{|k\rangle\}_{k=0}^{d-1}$ is the logical basis that defines the reference frame. The state of $d$-BS2 represents the measurement apparatus photons will confront. When the $d$-BS2 is inserted (removed), the $d$-MZI is closed (open), the detected probabilities at $\{D_0, \dots D_{d-1}\}$ are dependent (independent) on the $\{\theta_k\}_{k=0}^{d-1}$ configuration, thus revealing the $d$-path wave-like interference ($d$-mode particle-like quantization).

In order to probe the genuine wave-particle duality, the state of $d$-BS2 has to be determined after the photon has entered the $d$-MZI. In Fig. 1c, we adopt a modified version of the quantum-controlled delayed-choice experiment, recently proposed by Ionicioiu and Terno[42] and implemented in several double-path experiments[12–15]. By introducing a quantum-controlled BS that is in a coherent superposition of presence and absence, it represents a controllable experiment platform that can reveal wave or particle character, and their intermediate character[42]. In our multipath quantum delayed-choice scheme, the state of a general $d$-dimensional Hadamard operator $\hat{H}_d$ is coherently entangled with the control qubit of $|\psi\rangle_C$. The Hadamard $\hat{H}_d$ is implemented by a balanced $d$-BS, with elements $h_{i,j}^{(d)} = \frac{1}{\sqrt{d}}(-1)^{i\odot j}$, where $i\odot j$ denotes the bitwise dot product of the binary representations of $i$ and $j$. Note that in general the implementation of $d$-dimensional controllable Hadamard

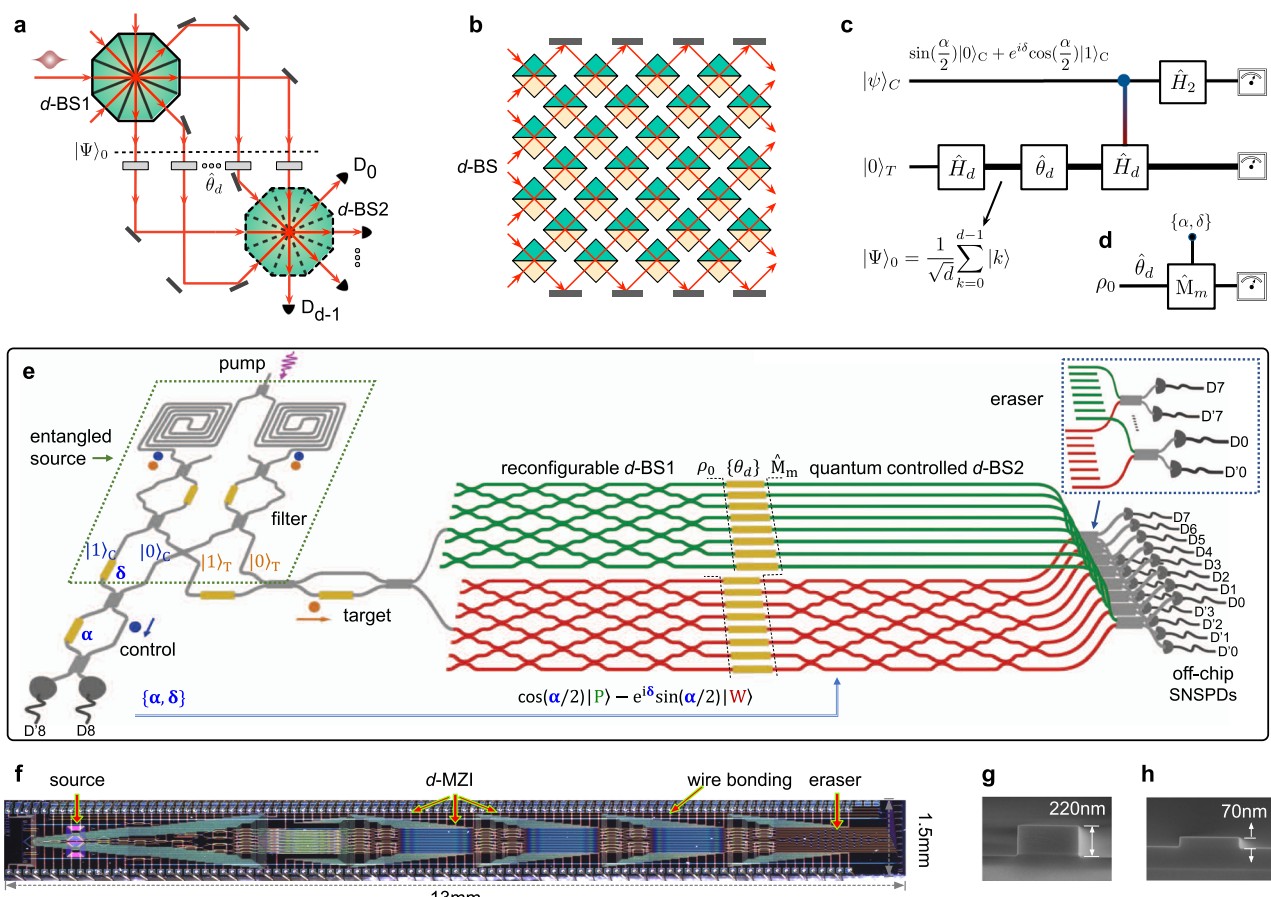

**Fig. 1 A quantum delayed-choice multipath experiment on a large-scale silicon-integrated quantum nanophotonic chip. a** Diagram for a *d*-path MZI, including two *d*-mode beamsplitters (*d*-BS1, *d*-BS2), and a phase array $\{\theta_d\}$ in the *d* paths. The presence or absence of *d*-BS2 (dashed) allows the measurement of wave or particle nature in the delayed-choice way. **b** An example of *d*-BS realization by nesting conventional bulk-optic BSs. The *d*-BSs can implement the generalized Hadamard operator $\hat{H}_d$. **c, d** Quantum circuit representation. The state of $\hat{H}_d$ operator is entangled with the state of a control qubit $|\psi\rangle_C$. Our target is to quantify the wave-particle dual nature of a maximal coherent state $\rho_0 = |\Psi\rangle_{00}\langle\Psi|$ by the delayed choice of measurement $\hat{M}_m$. $\hat{M}_m$ represents the *d*-mode particle or interference measurement (*m* is the measurement setting). **e** Simplified schematic, and **f** optical microscope image of the chip. The chip monolithically integrates an entangled photon-pair source, a *d*-path MZI, a quantum-controlled *d*-BS, and a *d*-mode eraser (note detectors are off chip). A pair of path-entangled photons (orange signal, blue idler photons) are generated in the integrated SFWM sources (dashed box). The target photon is sent through the *d*-MZI, undergoing either a *d*-mode wave process (bottom red circuits) or *d*-mode particle process (upper green circuits). If the control photon takes $|0\rangle$, the target undergoes the particle-process; if the control takes $|1\rangle$, the target undergoes the wave process; if the control is in a superposition state, the target undergoes the two processes coherently. A *d*-mode eraser ensures quantum indistinguishability between the two processes, see the zoom-in view in the inset. These result in the generalized quantum-controlled $\hat{H}_d$ or *d*-BS2. That being said, either wave or particle measurement $\hat{M}_m$ the photon takes is decided by the state of *d*-BS2, crucially, that is entangled with the control photon $\{\alpha, \delta\}$. Arbitrary $\{\theta_d, \alpha, \delta\}$ can be chosen on the chip. The integrated *d*-BSs are fully reconfigurable. The chip integrates 95 phase-shifters (golden lines), 158 beamsplitters, 58 waveguide crossers, 42 grating couplers, and 2 SFWM sources, in total 355 components (partially shown for clarity), among one of the largest quantum photonic devices. Photons are ultimately coupled off chip for detection by an array of fiber-coupled superconducting nanowire single-photon detectors $\{D_0, D'_0, ... D_8, D'_8\}$. **g, h** Scanning electron microscope (SEM) images for silicon photonic waveguides with 220 nm full etching and 70 nm shallow etching.

operation however is highly challenging in the delayed-choice experiment[10,11]. This is because of the difficulty of actively and rapidly operating the *d*-BS2—simultaneously operating the entire $(d^2 - d)/2$ array of 2-BSs, and thus quickly reconfiguring the whole *d*-MZI to an either open or closed state. We adopted a similar scheme as the double-path quantum delayed-choice experiments[12,13]. The two multimode complementary measurements performed on a target photon are determined by the state of another control photon, in which all operations only require passive optical components without any active operation of the whole *d*-BS2 and *d*-MZI. The key idea is to create a coherent entanglement between a control qubit and the state of *d*-BS2.

We devise a large-scale silicon-integrated quantum nanophotonic device for the implementation of the delayed-choice *d*-path

interferometric experiment, that features high phase stability and scalability[41]. Figure 1e illustrates a simplified diagram of the device to implement the circuit in Fig. 1c. Our task is to test the wave-particle dual nature of the target photon $\rho_0$ by choosing measurement apparatus $\hat{M}_m$ (Fig. 1d). The device includes four parts: an entangled photon-pair source, a *d*-path MZI, a quantum-controlled *d*-BS, and a *d*-mode eraser. As shown by Fig. 1e, all these parts are monolithically integrated on a single silicon chip. We prepare a maximally entangled Bell state $(|0\rangle_C|0\rangle_T + |1\rangle_C|1\rangle_T)/\sqrt{2}$ in two integrated spontaneous four-wave mixing (SFWM) sources[35], where $|0\rangle_{C,T}$ and $|1\rangle_{C,T}$ are path-encoded logical states of the control and target photons. The target photon undergoes a $\hat{H}_d$ transformation by the *d*-BS1, and a

phase operator $\hat{\theta}_d$ by the phase array. Depending on the control state of $|0\rangle_C$ or $|1\rangle_C$, the target photon coherently evolves either as a wave or particle, resulting in a state-process entanglement:

$$\frac{1}{\sqrt{2}}(|0\rangle_C|P\rangle_T + |1\rangle_C|W\rangle_T), \quad (1)$$

$$|P\rangle_T = \frac{1}{\sqrt{d}}\sum_{m=0}^{d-1} e^{i\theta_m}|m\rangle_T, \quad |W\rangle_T = \frac{1}{\sqrt{d}}\sum_{m=0}^{d-1}\sum_{k=0}^{d-1} h_{mk}^{(d)} e^{i\theta_k}|m\rangle_T, \quad (2)$$

where $|P\rangle$, $|W\rangle$ represent the states taking the particle and wave processes, respectively. The two processes are implemented by two distinct physical waveguide circuits, with $d$-BS2 (red circuits) and without $d$-BS2 (green circuits) (Fig. 1e). Which process the target photon experiences is coherently entangled with the state of control photon. If the control takes the state $|0\rangle$, the target undergoes the $d$-mode particle-process, revealing particle nature; if the control takes state $|1\rangle$, the target undergoes the $d$-mode wave process, exhibiting wave character; if the control photon is in a superposition state, it can reveal intermediate particle-wave characters. Importantly, the which-process information is erased at a $d$-mode quantum eraser (Fig. 1e), ensuring quantum mechanical indistinguishability between the wave and particle processes. These result in the realization of generalized quantum-controlled $\hat{H}_d$ operation or $d$-BS2. This state-process entanglement approach has been adopted for implementing double-path delayed-choice experiments[13] and for quantum simulations[43]. See Supplementary Note 1 and 3 for details.

We operate the control photon state as $\sin\frac{\alpha}{2}|0\rangle_C + e^{i\delta}\cos\frac{\alpha}{2}|1\rangle_C$, where $\{\alpha, \delta\}$ represents the angles of $\{\sigma_y, \sigma_z\}$ Pauli rotations, and selected the events when the detector $D'_8$ is clicked (Fig. 1e). Owing to the presence of entanglement between the control photon and the quantum state of $d$-BS2, the $d$-BS2 is thus in a superposition of presence and absence as $(\cos\frac{\alpha}{2}\hat{I} - ie^{i\delta}\sin\frac{\alpha}{2}\hat{H}_d)$. Note the first term represents the measurement of particle character, while the second term represents the measurement of wave character. Figure 1d represents the framework of delayed choice of measurement $\hat{M}_m = |m\rangle\langle m|$ performing on the target photon $\rho_0$, where $|m\rangle$ forms the wave-particle measurement basis:

$$|m\rangle = \frac{1}{\sqrt{N_d}}\sum_{k=0}^{d-1}\left(\Delta_{(m-k)}\cos\frac{\alpha}{2} + i\frac{e^{-i\delta}}{\sqrt{d}}(-1)^{m\odot k}\sin\frac{\alpha}{2}\right)|k\rangle, \quad (3)$$

where $m = 0 \ldots, d-1$ is measurement settings; $\Delta_x$ is the Kronecker delta function; $N_d$ is a normalization coefficient. See details in Supplementary Note 4. The probability of obtaining the $m$-th measurement outcome is quantified by $\mathrm{Tr}[\hat{M}_m\rho']$, where $\rho'$ is the state after the $\hat{\theta}_d$. In the experiment, we measured two-photon coincidences between a control port and any one of the $d$ target ports. The probabilities were then calculated by normalizing over the $d$ target ports.

The choice of measurement apparatus $\hat{M}_m$—determined by the $\{\alpha, \delta\}$ state of the control photon in a delayed manner, allows us to observe the $d$-path wave-particle transition and to test the $d$-path duality relation of the target photon. In Eq. (3), $\alpha$ refers to the amplitude of wave and particle properties, and the inherent $\delta$ phase identifies the genuine quantum particle-wave superposition. If $\alpha = 0$, $d$-BS2 is in the off-state and the $\hat{M}_m$ discloses the particle-nature. Hence, the photon registers each of the detectors (Fig. 1e) with a probability of $1/d$, leading to the observation of $d$-mode quantized distributions. If $\alpha = \pi$, $d$-BS2 is in the on-state and $\hat{M}_m$ reveals the wave nature. In this regard, the probability of detecting the photon relies on $\{\theta_d\}$, building up $d$-path wave interference patterns. If $0 < \alpha < \pi$, $d$-BS2 is in a superposition of the on- and off-state and it thus allows the observation of wave-particle nature simultaneously. We remark that the choice of $\hat{M}_m$

and the dual property of the target photon remain undetermined, until the control photon has been detected. This is because of the essence of entanglement that information is non-locally shared between the two photons.

Our quantum chip is designed for $d$-path ($d \leq 8$) experiments, in which the number of paths and mode number of $d$-BSs can be reconfigured. The $d$-BSs are formed by a squared mesh of 2-BSs (each is a 2-path MZI for full reconfigurability). The chip integrates 95 phase-shifters that are individually addressed and electronically driven. A telecom-band laser was used to pump two integrated SFWM sources and generate a pair of entangled photons. The signal photon is regarded as the target photon, while the idler photon is regarded as the control. The two photons were ultimately routed off the chip for detection by superconducting nanowire single-photon detectors $\{D_i, D'_i\}$, $i = 0, \ldots 8$ (Fig. 1e). The fabricated devices and waveguides are shown in Fig. 1f–h. See Methods and Supplementary Note 1 for more details of device fabrication and setup.

**Ruling out high-order interference.** Born's rule implies that multipath interference consists of all possible combinations of mutual interference. The possible presence of high-order interference could mask the test of wave-particle duality in the $d$-path interferometric experiment. Prior to testing the multipath wave-particle duality, we first rule out the presence of high-order interference. As an example, we implemented a four-path interference experiment. We measured the normalized Sorkin parameter $\kappa$, a ratio of high-order interference to second-order interference[23–25], and obtained the tight bound of $-0.0031 \pm 0.0047$ for the fourth-order interference (Fig. 2). Our experimental results confirm the absence of high-order interference,

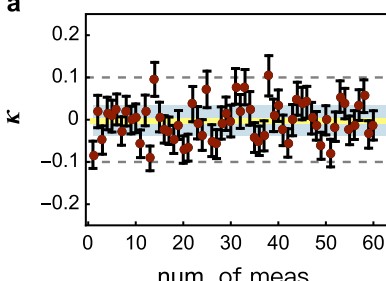

**a**

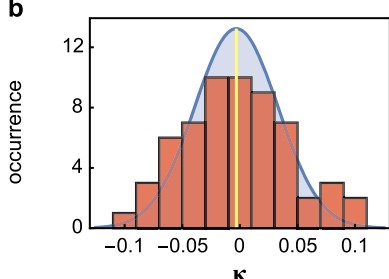

**b**

**Fig. 2 Measurement of high-order interference in the $d$-path interferometer. a** Measured $\kappa$ values for the fourth-order interference that is normalized to the second-order interference. In total, 60 measurements (red points) are performed independently. Error bars ($\pm\sigma$) are estimated from photon Poissonian statistics. The yellow line denotes the mean value of $\kappa$, and the shaded regime shows one standard deviation for all $\kappa$. **b** Histogram of all measured $\kappa$ values. The shaded regime shows a fitted Gaussian profile distribution. All data are measured at the prime maxima of the complete wave interference fringes. The measured tight bound of $\kappa = -0.0031 \pm 0.0047$ allows us to rule out the presence of high-order interference.

within an accuracy of $10^{-3}$ of the bound, which is comparable to the most precise result obtained so far in ref. [44]. Main experimental errors come from the non-perfect opening/closure of paths and thermal crosstalk between paths, as well as background photon noises. See Supplementary Note 2 for more measurement details.

**Multipath wave-particle transitions.** Figure 3 reports experimental results for $d$-path wave-particle transitions. Here, probability distributions for the $\hat{M}_0$-measurement are plotted, corresponding to the detection of the target photon at the ports of $\{D_0, D_0'\}$. Measurement results for other ports $\{D_i\}_{i=1}^{d-1}$ are provided in Supplementary Fig. 4. In our measurement, the phase $\theta_k$ was chosen as $k(\theta - \pi)$, where $\theta \in [0, 2\pi]$, but an arbitrary $\theta_k$ phase can be set. We obtained the continuous transition of $d$-path duality, from the full-particle nature at $\alpha = 0$ to the full-wave nature at $\alpha = \pi$, in two different scenarios: classical mixture (Fig. 3a–e) and quantum superposition (Fig. 3f–j). Classical wave-particle mixture represents the state $(\cos^2 \frac{\alpha}{2} |P\rangle\langle P| + \sin^2 \frac{\alpha}{2} |W\rangle\langle W|)$, while quantum wave-particle superposition represents the state $(\cos \frac{\alpha}{2} |P\rangle - ie^{i\delta} \sin \frac{\alpha}{2} |W\rangle)/\sqrt{N}$, where $N$ is a normalization coefficient, denoting the ultimate states after the quantum erasure. Details are given in Supplementary Note 3.

In the case of 2-path classical mixture, Fig. 3a shows a sinusoidal interference fringe at $\alpha = \pi$ representing the full-wave nature, and the detection probability approaches nearly 1/2 at $\alpha = 0$ representing the full-particle nature. The observation of 2-path wave-particle transition is consistent with the results in refs. [13–15]. In contrast, in the $d$-path experiments (Fig. 3b, c), at $\alpha = \pi$ we observed interference patterns that feature sharper distributions with an increment of $d$, confirming the $d$-path wave nature; at $\alpha = 0$ we observed a 1/4 (1/8) probability for $d = 4$ (8), confirming the $d$-mode particle nature; for $0 < \alpha < \pi$, intermediate particle-wave behaviors were revealed. The results for $\alpha = \{0, \pi\}$ are replotted in Fig. 3d, e, which are expected in classical optical multi-slit interference.

We now report the unique feature of quantum wave-particle superposition in $d$-path experiments. In Fig. 3f–h, the probability distributions represent asymmetry with respect to $\theta = \pi$, while the distributions for classical mixture in Fig. 3a–c remain symmetric. The extraordinary asymmetry comes from quantum interference between the wave and particle properties (Supplementary Eq. 14). Only when choosing the full-particle ($\alpha = 0$) or full-wave ($\alpha = \pi$) point, the distributions for classical (Fig. 3d, e) and quantum cases (Fig. 3i, j) are in agreement. When $\alpha \neq \{0, \pi\}$, the quantum distributions are remarkably distinct from the classical ones. Quantum distributions however tend to be less asymmetric for high $d$-path interference, becoming more classical (see analysis in Supplementary Note 5). Figure 3k–m shows quantum interference of multipath wave and multimode particle properties regarding the inherent phase $\delta$. We set $\alpha = 3\pi/2$ as an example (it works as well for $\pi/2$) that corresponds to the maximal wave-particle superposition. The $\delta$-dependence of distributions confirms the genuine quantum wave-particle superposition, while $\delta$-variation is absent in the case of classical mixture (see the explicit forms in Supplementary Eqs. 14 and 17). It is notable that by controlling the $\delta$ phase, the quantum interference of the wave and particle properties can be steered. For example, in the 2-path case, Fig. 3n shows constructive interference for $\delta = 0$ and destructive interference for $\delta = \frac{\pi}{2}$. In the case of 4-path interference (Fig. 3o), more wave-like characters appear when $\delta$ is set as $\frac{\pi}{2}$.

All measurements in Fig. 3 were performed in the computational basis $\{|0\rangle, |1\rangle\}$, which are well in agreement with theoretical predictions. We also estimated high-level classical fidelities (see

definition in Fig. 3's caption) of $0.998 \pm 0.001$ for 2$d$-path, $0.991 \pm 0.003$ for 4$d$-path, and $0.980 \pm 0.007$ for 8$d$-path experiments, respectively. Since entanglement is playing an enabling role in the delayed-choice measurement of $d$-path wave-particle duality, we repeated the experiments in the complementary basis $\{|+\rangle, |-\rangle\}$ to verify the presence of coherent entanglement, where $|\pm\rangle$ denotes $(|0\rangle \pm |1\rangle)/\sqrt{2}$. Our circuit allows any local qubit rotation by the MZIs together with posterior phase-shifters (Fig. 1e). In particular, local Pauli $\sigma_x \otimes \sigma_x$ operations on the Bell state were implemented to perform measurements in the $\{|+\rangle, |-\rangle\}$ basis. We again obtained coherent wave-particle transitions with high fidelities in the complementary basis, as shown in Supplementary Fig. 5. Moreover, to exclude the presence of local hidden variables that may contribute the delayed-choice measurement, we verified entanglement by both performing quantum state tomography of the entangled state, and demonstrating the violation of the Bell-CHSH (Clauser–Horne–Shimony–Holt) type inequality[45]. The quantum state fidelity of $0.962 \pm 0.002$ was obtained (Supplementary Fig. 3). And the Bell value of $2.75 \pm 0.04$ was measured, which violates the classical bound by $18.8\sigma$, confirming the existence of strong entanglement through the device.

**Generalized multipath wave-particle duality relation.** We next report experimental results of a generalized multipath wave-particle duality relation in the delayed-choice interferometer. It is of fundamental interest to develop a general framework to describe the multipath duality and to quantify wave and particle properties[17,27–30]. The conventional visibility defined as the contrast of 2-path interference fringe, fails to be a good wave measure of $d$-path ($d > 2$) interference[26]; however, quantum coherence is believed to be a good quantifier[27–30]. We adopt the $l_1$-norm coherence ($\widetilde{C}_{l1} = \sum_{i\neq j} |\rho_{ij}|$) proposed by Streltsov et al. as a wave measure[34]. Moreover, the capability of distinguishing which path the photon taken represents the $d$-path distinguishability. The formation of path-distinguishability for the 2-path case can be generalized to the $d$-path case[27,28,46]. The normalized coherence $C_d$ and path-distinguishability $D_d$ that we have used are given as:

$$C_d = \frac{1}{d-1} \sum_{i\neq j} \left| \rho_{ij} \right|, \tag{4}$$

$$D_d = \sqrt{1 - \left( \frac{1}{d-1} \sum_{i\neq j} \sqrt{\rho_{ii}\rho_{jj}} \right)^2}, \tag{5}$$

where $\rho$ represents the state for entire system having the target photon and measurement, as photon displays particle or wave nature is dependent on the measurement apparatus. The off-diagonal elements $\rho_{ij}$ determine the $d$-path wave interference, while the diagonal elements $\rho_{ii}$ determine the distinguishability of $d$-mode path-information. The explicit forms of $\rho$ for $d$-path classical and quantum experiments are given in Supplementary Note 4. Bohr's multipath duality rule is thus quantitatively generalized as[46]:

$$C_d^2 + D_d^2 \leqslant 1, \tag{6}$$

which saturates for the pure state, i.e., wave-particle superposition state. Proof is provided in Supplementary Note 4. The definitions of $C_d$ and $D_d$ well meet Dürr's criteria[27], and importantly, they represent macrovariables that capture the global features of the $d$-path interferometric system. Note when $d = 2$, the generalized duality relation in Eq. (6) reduces to the fundamental statement by Jaeger et al.[17].

The $C_d$ and $D_d$ were measured in the context of delayed choice of measurement $\hat{M}_0$ under the control of $\{\alpha, \delta\}$. It is notable that

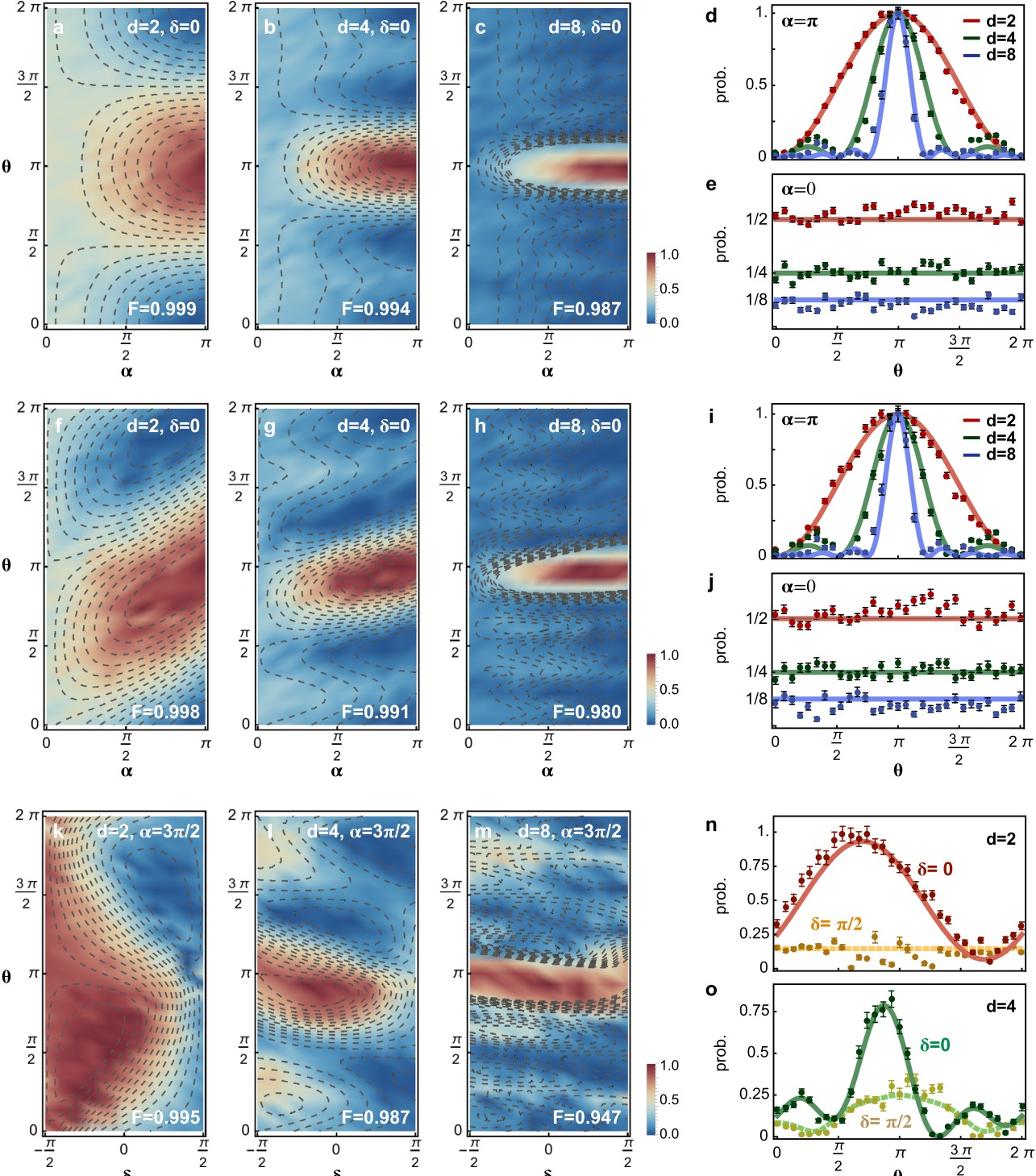

**Fig. 3 Experimental observations of multipath wave-particle transition in the delayed-choice experiment.** Measured transitions between particle and wave properties in several different scenarios: **a–c** classical mixture; **f–h**, quantum superposition; and **k–m**, intrinsic coherent quantum superposition. They are quantified by the probability distributions (normalized coincidences) for different $\{\alpha, \delta\}$ of the control and $\{\theta_d\}$ of the target ($\theta_k = k(\theta - \pi)$) was chosen, in the 2-, 4-, and 8-path experiments. Density distributions (colored) represent experimental data, while contour lines (dashed) represent theoretical results. The $F$ denotes the classical fidelity $\sum_i \sqrt{p_i q_i}$ summing over the whole space of $(\theta, \alpha)$ or $(\theta, \delta)$, where $p_i$ and $q_i$ are the measured and theoretical probabilities, respectively. High fidelities are obtained for all measurements. Results in **a–c** are consistent with classical optical multi-slit interference. The asymmetry of transition patterns in the quantum case **f–h** stem from quantum interference between wave and particle properties. Intrinsic coherent quantum superposition represents the intermediate particle-wave character, corresponding to the maximal wave-particle superposition, when $\alpha = 3\pi/2$ or $\pi/2$. The $\delta$-dependence of interference patterns in **k–m**, as an example measuring at $\alpha = 3\pi/2$, confirms the existence of genuine wave-particle superposition. **d**, **e** Classical fringes, and **i**, **j** quantum fringes for the full-wave case at $\alpha = \pi$, and full-particle case at $\alpha = 0$, for $d = 2$, 4, and 8. The interference fringe becomes sharper for $d$-path interference; the multimode quantization results in a $1/d$ probability at the outport. Only when $\alpha = \{0, \pi\}$, the classical and quantum fringes agree with each other. Examples of $\delta$-dependence interference fringes for $\delta = \{0, \pi/2\}$ and $\alpha = 3\pi/2$ in the **n** 2-path, and **o** 4-path experiments. The construction or destruction of interference appears by controlling the internal $\delta$ phase. Points represent experimental data, while lines represent theoretical values. All error bars ($\pm 3\sigma$) are estimated from photon Poissonian statistics.

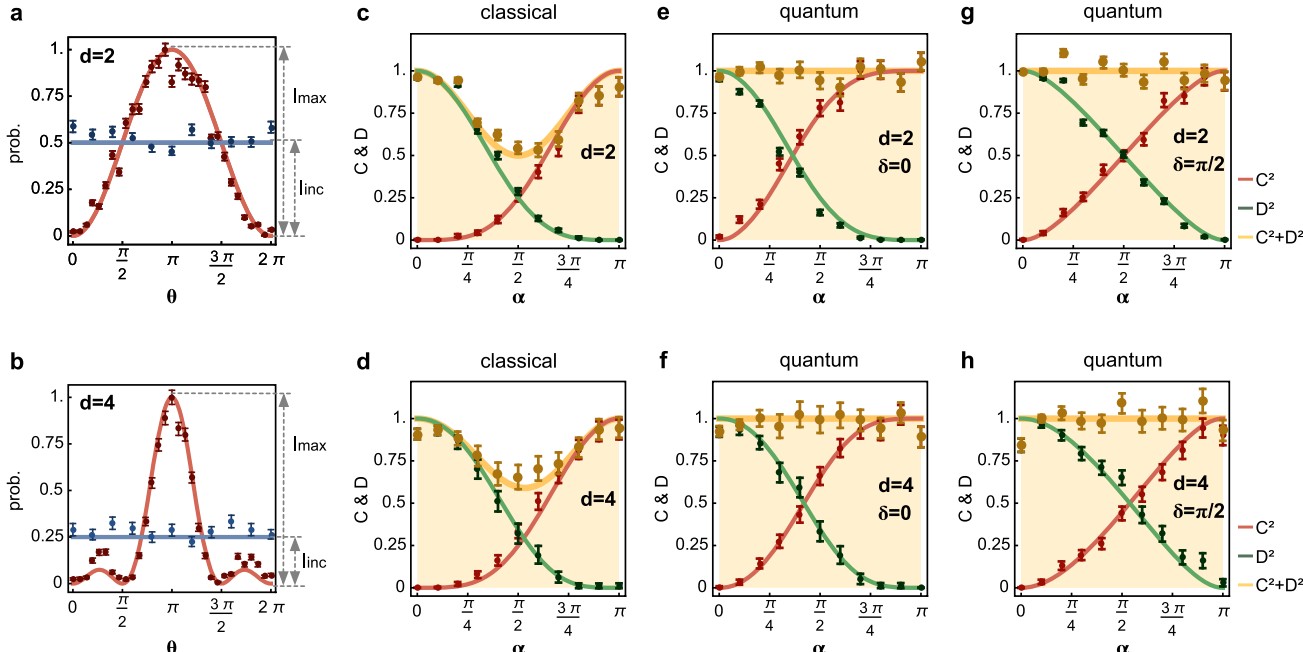

**Fig. 4 Experimental results of generalized multipath wave-particle duality relation in the delayed-choice experiment.** Measurement of generalized visibility ($\mathcal{V}_d$) that is equivalent to the normalized $l_1$-norm coherence ($\mathcal{C}_d$) for **a** 2-path and **b** 4-path full-wave interference ($\alpha = \pi$). The $\mathcal{V}_d$ is determined by the disparity of the primary maxima $I_{max}$ and incoherent term $I_{inc}$, both directly read out from the $d$-path interference fringes, with no need to probe the full density matrix. $I_{inc}$ was obtained by recording the probability of $i$-outcomes only having the $i$-path open, and the mean of measured $I_{inc}$ was used to estimate the $\mathcal{V}_d$. Measurement of generalized duality relation in the classical mixture scenario: **c** 2-path and **d** 4-path experiments; in the genuine quantum-superposition scenario: **e** 2-path, $\delta = 0$; **f** 4-path, $\delta = 0$; **g** 2-path, $\delta = \pi/2$; and **h** 4-path, $\delta = \pi/2$ experiments. The path-information $\mathcal{D}_d$ is measured in a posteriori way, by the delayed choice of measurement apparatus $\hat{M}_m$. The bound of $\mathcal{C}_d^2 + \mathcal{D}_d^2$ values are indicated within the orange-colored regimes. The generalization of duality relation $\mathcal{V}_d^2 + \mathcal{D}_d^2 \leq 1$ is thus confirmed in the $d$-path quantum-superposition experiment for different $\{d, \alpha, \delta\}$ configurations (**e–h**). In the classical-mixture case (**c, d**), the duality relation only holds at the full-wave and full-particle points at $\alpha = \{0, \pi\}$, independent on the setting of $\delta$. In all plots (**a–h**), points represent experimental data, while lines represent theoretical values. Error bars ($\pm\sigma$) are estimated from photon Poissonian statistics.

the measurement of which-path information $\mathcal{D}_d$ in our experiment is a posteriori one[11,13]. This is because of the fact that $\hat{M}_0$ enables the projection into the particle-wave superposition basis, that is delayed controlled from the full-wave to full-particle measurements. In order to measure the $\mathcal{C}_d$, we here adopt a newly derived general visibility $\mathcal{V}_d$ proposed by Qureshi et al.[32,46],

$$\mathcal{V}_d = \frac{1}{d-1}\frac{I_{max} - I_{inc}}{I_{inc}}. \quad (7)$$

Importantly, $\mathcal{V}_d$ is exactly equivalent to the normalized coherence, i.e., $\mathcal{C}_d = \mathcal{V}_d$, and it can be directly read out from interference patterns characterized by $\{I_{max}, I_{inc}\}$ (Fig. 4a, b), without knowledge of the density matrix of the system. We obtained the primary maxima $I_{max}$ of the interference fringes, and the incoherence contribution $I_{inc} = \frac{1}{d}\sum_{i=0}^{d-1}\rho_{ii}$. Take the case for $\alpha = \pi$ as examples, Fig. 4a, b shows the interference patterns for the 2-path and 4-path cases, from which the $\{I_{max}, I_{inc}\}$ values and thus the $\mathcal{V}_d$ values can be directly obtained (results for $d \in [2, 8]$ are reported in Supplementary Fig. 6). To get which-path information $\mathcal{D}_d$, we measured the diagonal elements $\rho_{ii}$, by recording the probability of $i$-outcomes only having the $i$-path open. We measured $\mathcal{V}_d$ and $\mathcal{D}_d$ for different $\{\alpha, \delta, d\}$. Figure 4c, e (d, f) reports the experimental results for the 2-path (4-path) case, when $\delta$ is set as 0. The results for $\delta = \frac{\pi}{2}$ are shown in Fig. 4g, h.

Figure 4c–h reports experimental results of the generalized multipath duality relation for both wave-particle quantum-superposition and classical-mixture cases. In the genuine wave-particle quantum-superposition case (Fig. 4e–h), we demonstrate that the generalized duality relation holds the tight bound of unity

as $\mathcal{C}_d^2 + \mathcal{D}_d^2 = 1$ in the different setting of $\{d, \alpha, \delta\}$. This is however not true for the classical-mixture case, where one cannot reach the upper bound and the equality does not saturate at $\alpha \neq \{0, \pi\}$, resulting in $\mathcal{C}_d^2 + \mathcal{D}_d^2 < 1$. The bound can only be approximated at $\alpha = \{0, \pi\}$ for full-particle or full-wave states, as shown in Fig. 4c, d. The quantitative results in Fig. 4 are consistent with the qualitative observations in Fig. 3a–j. The quantity of $\mathcal{L}_d = 1 - \mathcal{C}_d^2 - \mathcal{D}_d^2$ represents the missing information, which is due to a classical lack of knowledge about the quantum system. The $\mathcal{L}_d$ is directly related to the Tsallis 1/2-entropy as $\mathcal{L}_d = \frac{1}{d-1}\left(\frac{1}{4}(S_{1/2}(\rho) + 2)^2 - 1\right)$, where $S_{1/2}(\rho) = 2(\mathrm{Tr}(\rho^{1/2}) - 1)$ represents the Tsallis entropy. Note $\mathcal{L}_d$ is exactly the linear entropy if the $l_2$-norm coherence is used[28]. In Supplementary Note 5, we show how $\mathcal{L}_d$ scales for a large value of $d$.

## Discussion

Wave-particle duality is a fundamental feature of quantum physics. The sophisticated $d$-path interference patterns in fact contain rich information. For example, a direct quantification of the amount of coherence embedded in the $d$-path interference patterns is allowed by measuring the visibility $\mathcal{V}_d$ (Fig. 4a, b and Supplementary Fig. 6), without the need for explicitly referring to the $\rho$ of the system. We measured the $l_1$-norm coherence (see refs. [33,34]) $\widetilde{\mathcal{C}}_{l_1}$ for $d \in [2, 8]$, directly from the $d$-path full-wave interference fringes ($\alpha = \pi$). Figure 5a shows the results that identify a linear scale-up of $\widetilde{\mathcal{C}}_{l_1}$ information. In addition, the which-path information of $d$-outcomes is fundamentally non-deterministic and provides a way for quantum randomness

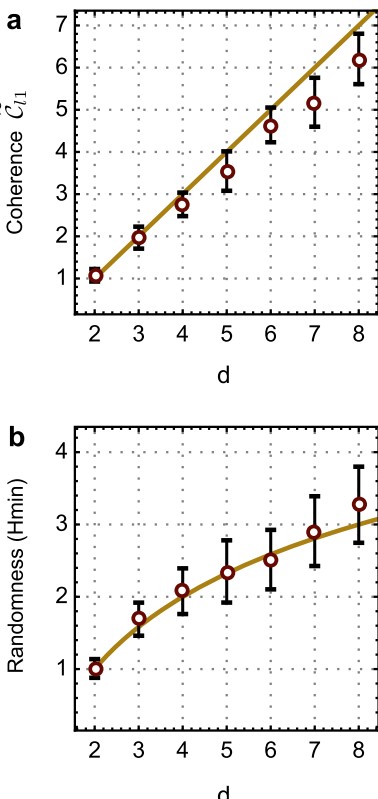

**Fig. 5 Characterization of multimode coherence and multimode quantization. a** Quantification of the $l_1$-norm quantum coherence $\tilde{\mathcal{C}}_{l_1}$ from the $d$-path full-wave interference patterns when $\alpha$ is chosen as $\pi$. The $\tilde{\mathcal{C}}_{l_1}$ values are estimated from the visibility by $\mathcal{V}_d(d-1)$, that are directly probed from the $d$-path interference patterns, without accessing the full density matrix. **b** Measurement of quantum randomness entropy $H_{\min}$ from the $d$-mode full-particle quantized distributions, when $\alpha$ is chosen as zero. More than one bit of randomness is obtained. The results in **a** and **b** are collected in the quantum wave-particle superposition measurement. Red points represent experimental data, while orange lines represent theoretical results. Error bars ($\pm\sigma$) are estimated from photon Poissonian statistics.

generation. Based on the $d$-mode quantizations, it is possible to generate more than one bit of randomness[35,47,48]. The randomness is quantified by min-entropy $H_{\min} = -\log_2 p_d$, where $p_d$ is the probability of correctly guessing which path the photons take. Figure 5b shows the measured $H_{\min}$ values, in the full-particle case ($\alpha = 0$), generating more than one bit of randomness.

Imperfect device fabrication and noisy operation ultimately contribute to the degradation of our experiment results, such as an accuracy of $10^{-3}$ of the bound for the high-order interference, and noises in the measurements of generalized wave-particle transition and duality relation. For example, photon leakages between neighboring paths in the $d$-MZI, owning to the presence of thermal crosstalks and imperfect extinction ratio of 2-MZIs (~30 dB), bring in noises in the measurement of high-order interference. Further optimizations of device fabrication and alleviations of crosstalk can improve the performance. Moreover, accidental counts may be induced from the undesirable SFWM process when the residual bright light propagates through the whole chip, i.e., the parts after the sources, though they are negligible in the measurement of coincidences. Such background noises can be further suppressed by the adoptions of pump rejection filter[49] and optical microresonator photon-pair source with a high coincidence-to-accidental ratio[50]. In our experiment, we relied on the assumptions that photons were faithfully

sampled, and the choice-maker and the observer were independent from each other; these assumptions could be further relaxed in future[51].

In conclusion, we have reported an experimental generalization of Bohr's duality relation in a delayed-choice experiment, on a large-scale silicon-integrated quantum optical chip. The multipath wave-particle transition and generalized duality relation have now been confirmed by the delayed choice of measurement performed on single photons. Such a multimode quantum system provides a versatile platform to study multimode quantum superposition and coherence—the most fundamental quantum properties and resources[33]. The direct probing of quantum coherence from interference distributions may allow the study of quantum processes and dynamics in complex quantum physical systems[52] and biological systems[53]. Going beyond the qubit-based quantum systems, highly controllable multidimensional quantum devices and systems that bases on the large-scale integrated quantum photonics platform are expected to continuously advance quantum information science and technologies[41,54,55].

## Methods

**Device fabrication.** The large-scale integrated quantum photonic device for the implementation of the multipath delayed-choice experiment was designed and fabricated on the silicon nanophotonics platform, a versatile system for photonic quantum information technologies. The quantum device was fabricated by the standard CMOS (complementary metal-oxide-semiconductor) processes. A layer of photoresist was the first spin on an 8-inch SOI (silicon-on-insulator) wafer with 220-nm-thick top silicon and 3 μm-thick buried oxide. The 248 nm DUV (deep ultraviolet) lithography was adopted to define the circuit patterns on the photoresist. Double inductively coupled plasma (ICP) etching processes were applied to transfer the patterns from the photoresist layer to the silicon layer, forming waveguides and circuits. Deep etching waveguides (see SEM image in Fig. 1g) with an etched depth of 220 nm were used for the SFWM photon sources, beamsplitters, and phase-shifters. Shallow etching waveguides (see SEM image in Fig. 1h) with an etched depth of 70 nm were used for the waveguide crossers and grating couplers. A SiO₂ layer of 1μm thickness was deposited on top of the SOI wafer by plasma-enhanced chemical vapor deposition (PECVD), working as an isolation layer between the waveguides and metal heaters to avoid potential optical losses. Then, a 10-nm-thick Ti glue layer, a 20-nm-thick TiN barrier layer, an 800-nm-thick AlCu layer, and a 20-nm-thick TiN anti-reflective layer, were consequently deposited by physical vapor deposition (PVD) and patterned by DUV lithography and etching process to form the electrode. A 50-nm-thick TiN layer for thermal-optical phase-shifters was deposited and also patterned by DUV lithography and etching process. Finally, another 1-μm-thick SiO₂ was deposited as the top cladding layer, and followed by the bonding pad opening process. More SEM images of the fabricated optical components and their characterizations are provided in Supplementary Note 1.

**Experimental setup.** Our experimental setup bases on off-the-shelf telecommunication instruments. A tunable continuous-wave laser (EXFO) central at a wavelength of 1550.11 nm was amplified to 40 mW by an erbium-doped fiber amplifier (EDFA, Pritel), and then fed in to the silicon-photonics quantum chip as a pump source for the SFWM nonlinear process. Before the chip, the polarization state of the pump light was optimized by a fiber polatisation controller in order to excite the transverse electric (TE) mode of silicon waveguides. A pair of path-coded entangled photons was on-chip generated by the two SFMW sources (Fig. 1), and further adopted for the implementation of multipath quantum delayed-choice experiment. We selected the signal photon at 1545.31 nm and idler photon at 1554.91 nm. After the chip, wavelength-division multiplexing (WDM) filters with a bandwidth of 1.1 nm and 200 GHz channel spacing were used to remove the residual pump light. Single photons were detected by an array of superconducting nanowire single-photon detectors (SNSPDs, Photonspot). A multichannel time interval analyzer (Swabian) was used to record two-fold photon coincidences. All of the 95 thermal-optical phase-shifters were accessed and controlled individually by multichannel electronics (Qontrol) with a 16-bits precision and KHz speed. The quantum chip was glued on a printed circuit board (PCB), and packaged by wire bonding (Fig. 1f). To achieve the full $2\pi$ operation of each phase-shifter, it requires about 40 mW power consumption. When all phase-shifters were on, total power consumption was about several watts. To ensure the stability of quantum operations and suppress thermal noises and thermal crosstalk, the chip was mounted on a Peltier-cell and a thermosink with liquid cooling in order to dissipate power rapidly and efficiently. A thermistor mounted on the chip together with a proportional integrative derivative controller was used to monitor and stabilize the temperature of the photonic device. See Supplementary Fig. 2 for more experimental details.

## Data availability

The data that support the findings of this study are available from the corresponding author upon reasonable request.

## Code availability

The codes that support the findings of this study are available from the corresponding author upon reasonable request.

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

## Acknowledgements

We thank T. Qureshi, P. Skrzypczyk, Y. Ding, and X. Yuan for useful discussions and comments. We acknowledge support from the National Key Research and Development (R&D) Program of China (nos 2019YFA0308702, 2018YFB1107205, 2016YFA0301302), the Natural Science Foundation of China (nos 61975001, 61590933, 61904196, 61675007, 11975026, 12075159), Beijing Natural Science Foundation (Z190005), and Key R&D Program of Guangdong Province (2018B030329001). S.F. acknowledges support from Shenzhen Institute for Quantum Science and Engineering, Southern University of Science and Technology (grant no. SIQSE202005), the Key Project of Beijing Municipal Commission of Education (grant no. KZ201810028042), and Academy for Multidisciplinary Studies, Capital Normal University. M.H. acknowledges support from the Austrian Science Fund (FWF) through the START project Y789-N27.

## Author contributions

J.W. conceived the project. X.C., Y.D., T.P., J.M., J.B, C.Z., T. D., and H.Y. built the setup and carried out the experiment. Y.Y., B.T., and Z.L. fabricated the device. X.C., Y.D., S.L., T.P., J.G., S.-M.F., M.H., and Q.H. performed the theoretical analysis. Q.H., Q.G., and J.W. managed the project. X.C., Y.D., S.L., and J.W. wrote the manuscript. All authors discussed the results and contributed to the manuscript.

## Competing interests

The authors declare no competing interests.
