## [Peer Review File · Nature Communications]

Reviewers' Comments:

Reviewer #1:

Remarks to the Author:

The authors experimentally probe the duality of waves and particles, i.e., the duality of coherence and which-way information, with photons in a delayed-choice interferometer with multiple (more than two) paths. Delayed-choice means that the experiment is not a priori designed to probe either wave or particle character, but that this choice is made only during the experiment. In this work specifically quantum entanglement is used to make this choice: entangled photon pairs are created, one photon is measured to collapse the other photon into a superposition of wave and particle character, which then probes the interferometer. This elegant trick has been established in Ref. 14 and other experiments with two paths, so the main technical advance provided by this work is its extension to multiple paths. However, this is not merely a technical advance but opens a route to directly study the wave-particle duality in multi-mode scenarios, which has been an area of intense scientific debate over the last two decades, as there is plenty of room for ambiguity on how to correctly assess wave and particle character in these settings.

The authors make use of very advanced Silicon integrated photonics technology to implement their photon source and the delayed-choice interferometer. After proving that a coherent superposition of wave and particle-character photons can be produced on their platform, they investigate the multi-mode wave-particle duality by demonstrating that a fundamental inequality between coherence and which-way information holds in all their settings and that it is always saturated for coherent superpositions, but not classical mixtures, of the character. This is the main scientific result of their work and, to my knowledge, the first fully-fledged experimental demonstration of the wave-particle duality relation in multi-mode settings.

In my opinion, this main result constitutes a substantial advance on the foundations of quantum physics, as it helps to clarify the aforementioned ambiguities and proves clearly that wave-particle duality carries over from double-path to multi-path settings (not unexpectedly, but such expectations need to be checked by experiments). The delayed-choice implementation illustrates also nicely the weirdness and the subtleties of the quantum world. Therefore, I fully recommend the topic and content of this work for publication in Nature Communications.

However, I find the quality of presentation in the current version of the manuscript insufficient. On the plus side, the structure of the paper makes sense to me, the abstract, introduction and conclusion are clear on what this work is about and the quality of all plots is good. On the other hand, the core of the manuscript suffers from a quite jargon-laden and dense writing style with sometimes confusing statements or missing important pieces of information. Here is a (non-exhaustive) list of issues, which should be improved:

- the terms "delayed determined/chosen" and "which-measurement" are quite unconventional, if not misleading. Using something like "determined/chosen in a delayed manner" and "which-path measurement" would improve the readability. "shifter-shifter" should be replaced by "phase-shifter".
- The paper starts with an explanation of the d-mode delayed-choice experiment (Fig. 1a), which is probably too steep a start for non-experts. I recommend explaining the basic two-mode setting first. In any case, it should be explained with an additional half-sentence that whether the photons take one or multiple paths to a given detector depends on the absence or presence of d-BS2
- It does not get fully clear from the text or Fig. 1 that the particle and the wave-interferometers are realised by two distinct physical waveguide circuits on the same chip, spanning a super-interferometer between the source and the erasing beam splitters
- It is difficult to connect the circuit in Fig. 1e with the quantum state equations given in Supplementary chapter 3. I think it would help to label the control and target modes in the figure and to indicate where exactly (after which beam splitter) the Bell state is prepared.
- It should be made clear after Eq. (3) that coincidences over a control port and the d target

output ports are measured (and not among target ports)

- The term “intrinsic coherent quantum-superposition” (Figure caption 3k-m) is utterly unclear. Why is this measurement made at $\alpha = 3/2 \pi$ and not at $\pi/2$, which is also half-way between particle and wave character?
- It should be explained how the basis rotation to $|+\rangle, |-\rangle$ is implemented

As far as I can tell, the most relevant literature on the main topic is cited, except the early experiment by Mei and Weitz (PRL 86, 559 (2001)), which shows an interesting non-trivial dynamics of interference visibility and which-path information in a four-path interferometer.

As a bonus, the authors also demonstrate the absence of higher-order interferences to a certain extent (Fig. 2 and associated text). This is interesting, but a bit disconnected from the rest of their work. Some motivation for this is given in the supplementary section 2. This should better be provided in the main text to make clear that if there was some higher-order interference, it could mask the testing of wave-particle duality in the multi-path setting. Some further comments on this part:

- What is Bohr’s rule? Do the authors mean Born’s rule? In any case, there is no postulate of interference arising from pairwise coherences. Instead, this pairwise interference follows from the postulate of Born’s rule, stating that the probability density is given by the modulus square of the wave function.
- The measurement of fourth-order κ could be compared to the most precise result obtained so far in this field (uncertainty of $1.8e-3$): Kauten et al. NJP 19, 033017 (2017) (<https://doi.org/10.1088/1367-2630/aa5d98>)
- In the discussion in Supplementary 2, residual photons transmitted through blocked paths are mentioned as a potential error source. However, a close inspection of Eq. (S2) shows that the contribution of this leakage to κ is exactly zero, as long as the residual transmission is constant and independent of the other paths. Cross-talk between the channels, on the other hand, is an issue which can lead to systematic shifts (not only to noise).

The authors end their main text with studying the scaling of particle-randomness with mode-number (Fig. 5b). The idea to use the random output path of spatially split photons for quantum random number generation is not particularly novel. The authors should cite some literature on this. One experiment comes to my mind (Gräfe et al. Nat. Photon. 8, 791 (2014); <https://doi.org/10.1038/NPHOTON.2014.204>), but there might be more recent or more relevant ones.

Some further technical aspects, which the authors should clarify/discuss in more detail:

- Is the calculation of the fidelity in Fig. 3 done by integrating/summing over the whole (θ , α/δ) phase space?
- The θ -average of the measured particle-like ($\alpha=0$) data points in Fig. 3e and j seem to deviate a bit from the theoretical lines. Can the authors explain this? Also, it does not become clear from Fig. 4 whether I_{inc} is extracted from the experimental data or from the theory lines.
- The unconverted pump light propagates from the spiral source through the whole chip before being filtered out by DWDMs. To which extent are photons also created by SFWM in the later parts of the chip and how much influence does that have on the results?
- How exactly is temperature stabilisation used to reduce the thermal cross-talk (Fig. caption S2)

With all these points fixed and with some overall polishing (I would recommend the authors to have a few non-expert colleagues read their paper to get some additional feedback on its accessibility), I think this work will be a worthy contribution to Nature Communications.

Some optional/minor issues:

- The definition of the visibility V_d is probably not ideally placed at Eq.(4), as it is introduced and discussed only two paragraphs later
- It seems like an unnecessary complication to discuss the unnormalized coherence in Fig. 5a instead of the normalised version (with $1/(d-1)$ prefactor), which is studied in the rest of the manuscript.
- Last sentence in the second-to-last paragraph on page 5 claims that destructive interference in Fig. 3n arises also for $\delta=-\pi/2$. However, this is not shown in the figure and should be constructive interference according to Fig. 3k
- Fig. S1c: ± 0.014 is probably the standard deviation of the distribution and not the error of the mean value
- Fig. caption S4, fourth line: "The $p=1$ data..." should be replaced by " $p=0$ "
- Last sentence of Supplementary 3.2: "reporting have level of classical fidelity" is unclear
- Eq. (S23): The Kronecker symbol should have the index $m-k$ (Minus is missing)
- Supplementary 4.2.3, Proof of Lemma: add "in the complex plane" to "...equation holds when all $z_i=\rho_k l^{(i)}$ have the same angle ..." (otherwise it might be unclear which angle is meant)

Reviewer #2:

Remarks to the Author:

The manuscript by Chen et al. describes a quantum delayed-choice experiment in the context of multi-path interferometry. The idea is to send a photon through an N-path interferometer and implement, or not, a generalised Hadamard operation contingent on the state of a control qubit. That Hadamard operation determines whether a which-path or an interference measurement is performed by the subsequent photon detection. Entanglement between the control and the path of the interfering qubit, along with choice of the control measurement basis, allows for continuous variation of the interfering qubit between the two extreme conditions.

The novelty of this work is the multipath interferometer, and the corresponding complementary relation that is introduced and tested. This is an interesting and significant contribution and, in my opinion, justifies its publication in Nature Communications. The experiment is well performed, the data is really nice, and analysed in significant detail. I particularly enjoyed the in-depth and clear discussion of the generalised complementarity relation (eq. 6 and the surrounding discussion) is interesting and elegant, and is a really nice contribution.

My one significant criticism of the manuscript is that the controlled-Hadamard operation is poorly explained. I found it frustrating that the authors almost implied that the circuit itself changed as a result of the control operation, which is clearly not the case. Rather, the circuit is passive, but the injection point into the integrated circuit (and thus also the effective operation) changes. This should be really clearly explained in the manuscript. Indeed, I will be stronger—it is unacceptable that this is not made crystal clear, as it is the heart of the paper. Of course, references to a similar idea implemented in two-path circuits is made. However, the reader should be able to understand the core mechanism of the scheme without having to trawl through previous works.

There is a bit of room for improvement in the writing, and some inconsistent use of notation. But this can probably be sorted out in the copyediting process.

In summary, I recommend this for publication, but I consider it essential that the authors take the space to give a clear explanation of the controlled generalised Hadamard operation.

Reviewer #1:

The authors experimentally probe the duality of waves and particles, i.e., the duality of coherence and which-way information, with photons in a delayed-choice interferometer with multiple (more than two) paths. Delayed-choice means that the experiment is not a priori designed to probe either wave or particle character, but that this choice is made only during the experiment. In this work specifically quantum entanglement is used to make this choice: entangled photon pairs are created, one photon is measured to collapse the other photon into a superposition of wave and particle character, which then probes the interferometer. This elegant trick has been established in Ref. 14 and other experiments with two paths, so the main technical advance provided by this work is its extension to multiple paths. However, this is not merely a technical advance but opens a route to directly study the wave-particle duality in multi-mode scenarios, which has been an area of intense scientific debate over the last two decades, as there is plenty of room for ambiguity on how to correctly assess wave and particle character in these settings.

The authors make use of very advanced Silicon integrated photonics technology to implement their photon source and the delayed-choice interferometer. After proving that a coherent superposition of wave and particle-character photons can be produced on their platform, they investigate the multi-mode wave-particle duality by demonstrating that a fundamental inequality between coherence and which-way information holds in all their settings and that it is always saturated for coherent superpositions, but not classical mixtures, of the character. This is the main scientific result of their work and, to my knowledge, the first fully-fledged experimental demonstration of the wave-particle duality relation in multi-mode settings.

In my opinion, this main result constitutes a substantial advance on the foundations of quantum physics, as it helps to clarify the aforementioned ambiguities and proves clearly that wave-particle duality carries over from double-path to multi-path settings (not unexpectedly, but such expectations need to be checked by experiments). The delayed-choice implementation illustrates also nicely the weirdness and the subtleties of the quantum world. Therefore, I fully recommend the topic and content of this work for publication in Nature Communications.

We are grateful to the reviewer for his/her time and expertise in reviewing our manuscript, and for expressing his/her enthusiasm for our work. The insightful comments from the reviewer have been helpful for allowing us to clarify and further strengthen our manuscript. We have fully addressed the comments point by point as below.

However, I find the quality of presentation in the current version of the manuscript insufficient. On the plus side, the structure of the paper makes sense to me, the abstract, introduction and conclusion are clear on what this work is about and the quality of all plots is good. On the other hand, the core of the manuscript suffers from a quite jargon-laden and dense writing style with sometimes confusing statements or missing important pieces of information. Here is a (non-exhaustive) list of issues, which should be improved:

- the terms “delayed determined/chosen” and “which-measurement” are quite unconventional, if not misleading. Using something like “determined/chosen in a delayed manner” and “which-path measurement” would improve the readability. “shifter-shifter” should be replaced by “phase-shifter”.

We thank the reviewer for pointing out the unclear descriptions. As the reviewer suggested, we have now replaced the term of “delayed determined/chosen” by “determined/chosen in a delayed manner”, in the revised manuscript. Regarding the term of “which-measurement”, it describes either which-path measurement or interference measurement performed on the photon to reveal its particle nature or wave nature. The term of “which-path measurement” refers to that which path the photon takes in the absence of d-BS2. In the revised manuscript, to avoid ambiguity we have now replace the statement of “which-measurement” by “measurement apparatus” or “wave or particle measurement” or “two complementary measurements”. We have corrected the typo of “shifter-shifter” by “phase-shifter”.

- The paper starts with an explanation of the d-mode delayed-choice experiment (Fig. 1a), which is probably too steep a start for non-experts. I recommend explaining the basic two-mode setting first. In any case, it should be explained with an additional half-sentence that whether the photons take one or multiple paths to a given detector depends on the absence or presence of d-BS2

We agree with the reviewer that starting from the well-known double-path (double-slit) experiment can make the representation much more smoothly. Therefore, we have added a new first paragraph in the Introduction part of the revised manuscript, and the following description in the second paragraph.

Page 1: “Whether the photons take one or multiple paths to a given detector depends on the absence or presence of the second d-mode beamsplitter.”

- It does not get fully clear from the text or Fig. 1 that the particle and the wave-interferometers are realised by two distinct physical waveguide circuits on the same chip, spanning a super-interferometer between the source and the erasing beam splitters.

We have now added detailed descriptions of our device in both main text and caption of Fig.1. All components shown in Fig.1e are integrated on a single silicon chip, including the entangled photon source, two layers of distinct physical waveguide circuits performing either d-mode particle or wave interferometers, as well as the erasing beam splitters that ensure quantum mechanical indistinguishability of the wave and particle layers.

Figure 1 caption: “The chip (yellow box) monolithically integrates an entangled photon-pair source, a d-path MZI, a quantum-controlled d-BS, and a d-mode eraser.” ... “The target photon is sent through the d-MZI, undergoing either a d-mode wave process (red circuits) or d-mode particle process (cyan circuits). If the control photon takes $|0\rangle$, the target undergoes the particle-process; if the control takes $|1\rangle$, the target undergoes the wave-process; if the control is in superposition state, the target undergoes the two processes coherently.”

Page 3: “As shown by Fig.1e, all these parts are monolithically integrated on a single silicon chip (indicated by the yellow box).”

Page 4: “The two processes are implemented by two distinct physical waveguide circuits, with d-BS2 (red circuits) and without d-BS2 (cyan circuits), see Fig.1e. Which process the target photon experiences is coherently entangled with the state of control photon. If the control takes the state $|0\rangle$, the target undergoes the d-mode particle process, revealing particle nature; if the control takes state $|1\rangle$, the target undergoes the d-mode wave-process, exhibiting wave character; if the control photon is in a superposition state, it can reveal intermediate particle-wave characters. Importantly, the which-process information is erased at a d-mode quantum eraser (see Fig.1e), ensuring quantum mechanical indistinguishability between the wave and particle processes. These result in the realisation of generalised quantum-controlled H_d operation or d-BS2.

- It is difficult to connect the circuit in Fig. 1e with the quantum state equations given in Supplementary chapter 3. I think it would help to label the control and target modes in the figure and to indicate where exactly (after which beam splitter) the Bell state is prepared.

We have now modified the Figure 1e to clarify where the entangled photons are produced, and the logical modes of the control ($|0\rangle_c, |1\rangle_c$) and target ($|0\rangle_t, |1\rangle_t$) photons have been labeled.

Figure 1 caption: “...in the integrated SFWM sources (dashed box)”

- It should be made clear after Eq. (3) that coincidences over a control port and the d target output ports are measured (and not among target ports)

We thank the reviewer for highlighting the necessity of measuring coincidences in our experiment. Indeed, the choice of wave or particle measurement on the target photon is dependent on the state of the control photon. This requires the joint coincidence measurement between the two photons.

In the paragraph after Eq. (3), we have now added one sentence:

“In our experiment, the probabilities for each measurement are calculated by normalizing two-fold coincidence counts over all outcomes between a control port and the d target ports.”

- The term “intrinsic coherent quantum-superposition” (Figure caption 3k-m) is utterly unclear. Why is this measurement made at $\alpha = 3/2 \pi$ and not at $\pi/2$, which is also half-way between particle and wave character? The referee is correct that the setting of phase $\alpha = 3/2 \pi$ or $\pi/2$ works exactly the same in our experiment, both carrying half-way between particle and wave character.

We have added this notion in Page 6, containing the following text:

Page 6: “We set $\alpha = 3\pi/2$ as an example (it works as well for $\pi/2$) that corresponds to the maximal wave-particle superposition.”

We have added the description of “intrinsic coherent quantum-superposition” in the caption of Figure3:

“Intrinsic coherent quantum-superposition represents the intermediate particle-wave character, corresponding to the maximal wave-particle superposition, when $\alpha = 3\pi/2$ or $\pi/2$. The delta-dependence of interference patterns in (k-m), as an example measuring at $\alpha = 3\pi/2$,...”

- It should be explained how the basis rotation to $|+\rangle, |-\rangle$ is implemented

We verified the multi-path wave-particle transitions in both the Z-basis and X-basis. In the latter case, we locally rotated the two qubit Bell state $|00\rangle + |11\rangle$ into $|++\rangle + |--\rangle$, and implemented the X-basis measurement of wave-particle transitions. The local Pauli-X rotation is on-chip implemented by a set of MZI together with a phase shifter, which enables performing any local rotation of single qubit state.

We have now added the detailed description in the revised manuscript:

Page 7: “Local Pauli \$\sigma_x \otimes \sigma_x\$ operations on the Bell state were implemented on-chip by the MZIs together with the posterior phase-shifters, as shown in Fig.1e.”

As far as I can tell, the most relevant literature on the main topic is cited, except the early experiment by Mei and Weitz (PRL 86, 559 (2001)), which shows an interesting non-trivial dynamics of interference visibility and which-path information in a four-path interferometer.

We thank the reviewer for pointing out the reference of PRL 86, 559 (2001), which now has been added as Ref.26.

As a bonus, the authors also demonstrate the absence of higher-order interferences to a certain extent (Fig. 2 and associated text). This is interesting, but a bit disconnected from the rest of their work. Some motivation for this is given in the supplementary section 2. This should better be provided in the main text to make clear that if there was some higher-order interference, it could mask the testing of wave-particle duality in the multi-path setting.

We are glad that the reviewer acknowledges the importance of ruling out high-order interference terms before the characterization of d-path wave-particle duality. As the referee suggested, to emphasize the necessity of such measurements and to smoothly connect this part with the duality measurements, we have now added its motivation in the revised manuscript.

Page 4: “The possible presence of high-order interference could mask the test of wave-particle duality in the d-path interferometric experiment. Prior to testing the multipath wave-particle duality, we first rule out...”

Some further comments on this part:

- *What is Bohr’s rule? Do the authors mean Born’s rule? In any case, there is no postulate of interference arising from pairwise coherences. Instead, this pairwise interference follows from the postulate of Born’s rule, stating that the probability density is given by the modulus square of the wave function.*

The referee is correct that in the part of high-order interference measurement, we refer to the “Born’s rule”. We have corrected this typo throughout the revised manuscript. We have also rephrased the sentence as:

Page 4: “Born’s rule implies that multipath interference consists of all possible combinations of mutual interference.”

- *The measurement of fourth-order kappa could be compared to the most precise result obtained so far in this field (uncertainty of $1.8e-3$): Kauten et al. NJP 19, 033017 (2017) (<https://doi.org/10.1088/1367-2630/aa5d98>)*

We thank the reviewer for pointing out the reference of NJP 19, 033017 (2017), which has been added as Ref.44.

We have also added a sentence as:

Page 4: “...which is comparable to the most precise result obtained so far in ref.[44]”

- *In the discussion in Supplementary 2, residual photons transmitted through blocked paths are mentioned as a potential error source. However, a close inspection of Eq. (S2) shows that the contribution of this leakage to kappa is exactly zero, as long as the residual transmission is constant and independent of the other paths. Cross-talk between the channels, on the other hand, is an issue which can lead to systematic shifts (not only to noise). Given an ideal device (i.e., d-MZI) with perfect on-off switching ratios, we could rule out the possibility of photon leakage across the paths. In reality, each of our 2d MZIs provides about 30db extinction ratio (see Supplementary Fig1.c). This means that we could not completely turn on/off paths in the current integrated optic device. There is about 0.1% possibility of photon leakage between neighboring paths, and such leakage could bring in noises and errors in the measurement of high-order interference.*

The Supplementary Eq. (S2) describes the fourth-order interference which contains three terms. The first term $P_{IV}(1234)$ refers to the case having all four paths open, contributing no leakage of photon within the four paths (but could be some degrees of leakage from the remaining four paths when reconfiguring our 8-path MZI). The second term of $P_{II}(ij)$ refers to the case that two of the paths are open while the others are blocked (noise could come from the other two paths). The third term of $P_I(i)$ refers to the case that only one of the paths is open while all other three paths are blocked (noise could come from the other three paths).

The referee is totally correct that thermal cross-talk is one of the main problems in current integrated quantum photonic platform, especially when the system scales up to the large-scale. In the measurement of high-order interference, the presence of cross-talk can indeed induce both noise as well as operation inaccuracy that could make experiment data deviate from theoretical ones. Please see more discussions in our reply to the comments concerning “temperature stabilization” as below.

We have added more discussions in Discussion section of the revised manuscript, and in Supplementary Note 2.

The authors end their main text with studying the scaling of particle-randomness with mode-number (Fig. 5b). The idea to use the random output path of spatially split photons for quantum random number generation is not particularly novel. The authors should cite some literature on this. One experiment comes to my mind (Gräfe et al. Nat. Photon. 8, 791 (2014); <https://doi.org/10.1038/NPHOTON.2014.204>), but there might be more recent or more relevant ones.

Yes, the generation of quantum random number by splitting the wavefunction of photons into two-path or d-path is not new. The reference of Nat. Photon. 8, 791 (2014), and a few other relevant references are cited now.

Some further technical aspects, which the authors should clarify/discuss in more detail:

- Is the calculation of the fidelity in Fig. 3 done by integrating/summing over the whole (theta, alpha/delta) phase space?

The answer is yes. We calculated the classical fidelity of the measured interference distributions by summing over the whole (theta, alpha) space in Fig.3 a-j, and over the whole space of (theta, delta) in Fig.3 k-o. We have added detail in the caption of Fig.3 and Supplementary Fig.5 to clarify the definition of classical fidelity.

- The theta-average of the measured particle-like (alpha=0) data points in Fig. 3e and j seem to deviate a bit from the theoretical lines. Can the authors explain this? Also, it does not become clear from Fig. 4 whether I_{inc} is extracted from the experimental data or from the theory lines.

The particle-like (alpha=0) data points in Fig. 3e and j were measured when the second d-BS is fully open. Any imperfection setting of the d-mode BS such as imperfect on-off switching ratio of the MZIs, as well as crosstalk between paths will result in interference. We have added explanations in the Discussion section.

I_{inc} in Fig.4 was extracted from experiments (blue points are the experimental data, and blue lines are the theoretical values). Indeed, I_{inc} corresponds to the summation of diagonal terms of the density matrix ρ_{ii} . We measured the diagonal elements, by recording the probability of i-outcomes only having the i-path open. In the revised manuscript, we have added this detail in the caption of Fig.4 and Supplementary Fig.6.

" I_{inc} was obtained by recording the probability of i-outcomes only having the i-path open, and the mean of measured I_{inc} was used to estimate the V_d .".....In all plots (a-h), points represent experimental data, while lines represent theoretical values."

- The unconverted pump light propagates from the spiral source through the whole chip before being filtered out by DWDMs. To which extent are photons also created by SFWM in the later parts of the chip and how much influence does that have on the results?

The reviewer is correct that photon-pairs can be created in the remaining parts of the chip after the spiral source. However, these photons had a nearly negligible contribution to all of our measurements of multipath wave-particle transitions and duality. As the referee pointed out before, we actually measured the two photon coincidences between the detector D8 and D0-D7. This coincidence measurement excludes the counting of the extra photons which are created after the spiral source.

One may argue that the photons created in the latter parts may bring in accidental coincidences. In order to suppress such accidental coincidences as much as possible, we used expanded waveguides up to a 5um-width to reduce the SFWM effect in some of the structures such the connecting and routing waveguides. And also note that the adoption of on-chip asymmetric MZIs (filters) can remove 50% of the residual pump, thus reducing the generation accidental coincidences. All these processes ensure small influence of our measurements. This was confirmed by the observation of high quantity two-photon interference (supplementary Fig.1b) and tomography data (supplementary Fig.3), as well as high fidelity wave-particle transitions in main text. In future, the uses of pump rejection filter (Nat. Photon. 12, 534-539(2018)) or optical resonator source (Nat. Phys. 16, 148-153(2020)) can help further suppress background noises in the whole process.

We have added more discussion in the Discussion section of the revised manuscript, and Supplementary Note1.

- How exactly is temperature stabilisation used to reduce the thermal cross-talk (Fig. caption S2)

The device is one of the largest silicon quantum photonic devices at the present, which integrates 95 thermo-optic phase-shifters in a single chip. To achieve full 2π operation of each phase-shifter, it requires about 40mW power consumption. When all phase-shifters are turned on, the total power consumption is about several watt. This would heat up the whole chip dramatically and induce strong crosstalk without temperature stabilization. Thus, in our experiment, the chip was mounted on a Peltier-cell and liquid cooling thermalsink which rapidly and efficiently dissipates power. And a thermistor mounted on the chip together with a proportional integrative derivative controller were used to keep the temperature of the photonic device stable.

We have now added the details in the Methods (experimental setup), and Supplementary Note1.

With all these points fixed and with some overall polishing (I would recommend the authors to have a few non-expert colleagues read their paper to get some additional feedback on its accessibility), I think this work will be a worthy contribution to Nature Communications.

We believe the referee's concerns have been all addressed here and also in the revised manuscript. We again thank the reviewer for his/her positive and constructive comments and for the opportunity to improve the presentation of our manuscript.

Some optional/minor issues:

- The definition of the visibility V_d is probably not ideally placed at Eq.(4), as it is introduced and discussed only two paragraphs later

As the referee suggested, we have now moved the definition of the visibility V_d to the later paragraph.

- It seems like an unnecessary complication to discuss the unnormalized coherence in Fig. 5a instead of the normalised version (with $1/(d-1)$ prefactor), which is studied in the rest of the manuscript.

The normalized coherence term C_d defined in Eq.(4) reflects its physical meaning that is exactly equivalent to the visibility V_d now defined in Eq.(7), and interestingly C_d can be directly measured in experiment as shown in Fig.4. Together with the path indistinguishability V_d , it allows us to test the generalized multipath duality relation. In contrast, the unnormalized coherence term C_{11} as shown in Fig.5a actually represents the well-known l_1 -norm coherence, introduced in ref. [PRL 113, 140401 (2014)]. Quantifying such l_1 -norm coherence in experiment is of practical significance in the field of resource theory of quantum coherence. And it was thought that the quantification of l_1 -norm coherence requires the characterization of the whole density matrix of the system. Figure 5a shows a direct experimental quantification of the l_1 -norm coherence, i.e, the unnormalized coherence, from the multipath wave interference patterns.

We however agree with the referee that this part of measurement of l_1 -norm coherence (unnormalized coherence) is not directly relevant to the main part of the multipath duality. In this part, together with the measurement of random numbers, we would like to give simple demonstrations of how the basic features of multimode quantum physics, such as coherence and quantization, can be adopted for quantum information applications. To make the main body of our work more focused, we thus have moved the whole Fig.5 parts into the Discussion section.

- Last sentence in the second-to-last paragraph on page 5 claims that destructive interference in Fig. 3n arises also for $\delta = -\pi/2$. However, this is not shown in the figure and should be constructive interference according to Fig. 3k

We thank the reviewer for pointing out the error. We have now corrected the descriptions in main text to consist with Fig.3n.

- Fig. S1c: ± 0.014 is probably the standard deviation of the distribution and not the error of the mean value
We thank the reviewer for correcting our statements of error bar. The error bar (± 0.014) in Fig. S1c indeed refers to the standard deviation of the distribution of measured visibilities. We have corrected those statements in Supplementary Note 1 and Supplementary Fig.1 caption.

- Fig. caption S4, fourth line: "The $p=1$ data..." should be replaced by " $p=0$ "

We thank the reviewer for pointing out this typo, which have now been corrected.

- Last sentence of Supplementary 3.2: "reporting have level of classical fidelity" is unclear

We thank the reviewer for pointing out our ambiguous expression, which have now been corrected.

- Eq. (S23): The Kronecker symbol should have the index $m-k$ (Minus is missing)

We thank the reviewer for pointing out this typo, which have now been corrected.

- Supplementary 4.2.3, Proof of Lemma: add "in the complex plane" to "...equation holds when all $z_i = \rho_k |e^{i\theta_k}|$ have the same angle ..." (otherwise it might be unclear which angle is meant)

We thank reviewer's suggestion and we have now added it into the Proof of Lemma as the referee suggested.

Reviewer #2:

The manuscript by Chen et al. describes a quantum delayed-choice experiment in the context of multi-path interferometry. The idea is to send a photon through an N-path interferometer and implement, or not, a generalised Hadamard operation contingent on the state of a control qubit. That Hadamard operation determines whether a which-path or an interference measurement is performed by the subsequent photon detection. Entanglement between the control and the path of the interfering qubit, along with choice of the control measurement basis, allows for continuous variation of the interfering qubit between the two extreme conditions.

The novelty of this work is the multipath interferometer, and the corresponding complementary relation that is introduced and tested. This is an interesting and significant contribution and, in my opinion, justifies its publication in Nature Communications. The experiment is well performed, the data is really nice, and analysed in significant detail. I particularly enjoyed the in-depth and clear discussion of the generalised complementarity relation (eq. 6 and the surrounding discussion) is interesting and elegant, and is a really nice contribution.

We thank the reviewer for his/her time and expertise in reviewing our work and providing insightful comments on the d-mode controlled-Hadamard operation, which we have now fully addressed in the revised manuscript. We thank the reviewer for his/her acknowledgement on the significance and interests of demonstrating the generalised complementarity relation in a multi-path interferometric experiment.

My one significant criticism of the manuscript is that the controlled-Hadamard operation is poorly explained. I found it frustrating that the authors almost implied that the circuit itself changed as a result of the control operation, which is clearly not the case. Rather, the circuit is passive, but the injection point into the integrated circuit (and thus also the effective operation) changes. This should be really clearly explained in the manuscript. Indeed, I will be stronger—it is unacceptable that this is not made crystal clear, as it is the heart of the paper. Of course, references to a similar idea implemented in two-path circuits is made. However, the reader should be able to understand the core mechanism of the scheme without having to trawl through previous works.

We thank the referee for insightfully pointing out the importance of clarifying how the generalised quantum-controlled Hadamard operation is implemented in our integrated device, which is indeed the core of our work. The referee is correct that our implementation of generalised quantum-controlled Hadamard operation is enabled by controlling the injection states of the control photon with the assistance of entanglement, instead of by actively and rapidly reconfiguring (opening or closing) the generalised d-mode beamsplitter on the target photon (that would be extremely challenging in experiment). We have clarified this in the revised manuscript. More than that, the reviewer's comment/suggestion can significantly highlight the technological challenge and our progress of realizing such generalised Hadamard operation, and we have now added it in our revised manuscript. The referee is totally correct that similar ideas of implementing delayed-choice experiment with quantum controlled two-mode Hadamard operation had been realized in the two-path systems, while our contribution and novelty in this work is the demonstration of generalized wave-particle duality in a multi-path delayed-choice experiment.

Following the referee's suggestion, we have added the following paragraphs in the revised manuscript:

Page 2: "Note that in general the implementation of d-dimensional controllable Hadamard operation however is highly challenging in the delayed-choice experiments [10,11]. This is because of the difficulty of actively and rapidly operating the d-BS2 – simultaneously operating the entire $(d^2 - d)/2$ array of 2-BSs, and thus quickly reconfiguring the whole d-MZI to an either open or closed state. We adopted a similar scheme as the double-path quantum delayed-choice experiments [12, 13]. The two multimode complementary measurements performed on a target photon are determined by the state of another control photon, in which all operations only require passive optical components without any active operation of the whole d-BS2 and d-MZI. The key idea is to create a coherent entanglement between a control qubit and the state of d-BS2."

Page 4: "The two processes are implemented by two distinct physical waveguide circuits, with d-BS2 (red circuits) and without d-BS2 (cyan circuits), see Fig.1e. Which process the target photon experiences is coherently entangled with the state of control photon. If the control takes the state $|0\rangle$, the target undergoes the d-mode particle process, revealing particle nature; if the control takes state $|1\rangle$, the target undergoes the d-mode wave-process, exhibiting wave character; if the control photon is in a superposition state, it can reveal intermediate particle-wave characters. Importantly, the which-process information is erased at a d-mode quantum eraser (see Fig.1e), ensuring quantum mechanical indistinguishability between the wave and particle processes. These result in the realisation of generalised quantum-controlled H_d operation or d-BS2."

Figure 1 caption: "The target photon is sent through the d-MZI, undergoing either a d-mode wave process (red circuits) or d-mode particle process (cyan circuits). If the control photon takes $|0\rangle$, the target undergoes the particle-process; if the control takes $|1\rangle$, the target undergoes the wave-process; if the control is in superposition state, the target undergoes the two processes coherently....These result in the generalised quantum-controlled Hadamard or d-BS2."

And we have also modified Fig.1e to further clarify the scheme.

There is a bit of room for improvement in the writing, and some inconsistent use of notation. But this can probably be sorted out in the copyediting process.

We have now checked through the manuscript, and improved the presentation of our work and corrected some inconsistent use of notations.

In summary, I recommend this for publication, but I consider it essential that the authors take the space to give a clear explanation of the controlled generalised Hadamard operation.

We have now fully addressed the referee's concerns in the revised manuscript, clarifying how we implemented the generalised quantum controlled Hadamard operation in our system. We again thank the reviewer for his/her positive and constructive comment/suggestion and for the opportunity to highlight an important technical point of our experiment.

Reviewers' Comments:

Reviewer #2:

Remarks to the Author:

I have been asked to review the revised manuscript and comment on the response of the authors to both referees 1 and 2.

First let me commend the authors on addressing the comments in detail. The manuscript is almost ready for acceptance. I list the remaining issues to be addressed before acceptance. None of these is a fundamental science problem. They are about how things are described in the manuscript.

1. One **significant element** of the text still needs fixing. Although the authors have made it much clearer that the choice of which-path or interference experiments is not actively switched on the timescale of the photon propagation, and have generally improved the discussion, I believe the manuscript still lacks a crystal-clear statement of **exactly** how this entanglement works out in practice.

Let me explain further. Let's consider just the extreme cases of the which-path or interference experiments, and ignore the in-between case. Obviously "which path" or "interference" is selected by getting one or other outcome in a measurement of the control qubit in the relevant basis. In turn, these outcomes project the target state into taking one mode or another early in the circuit (red arrow or cyan arrow, before d-BS1, in fig 1e). What is still not crystal clear is this: where do those two modes go? Do they go into two different physical copies of circuit, one with the d-Hadamard d-BS2 and one without? Do they go into two different modes of the fan-out? If so, how does this make a difference? Presumably they don't go into the same input mode of d-BS1 as this breaks unitarity. What I believe really needs to be explained in laborious detail is what actually happens specifically to each of these modes—where they go and why—so that the operation of the right-hand two thirds of the apparatus can be fully understood. At the moment, I feel that many readers will find this unclear. In the current version of figure 1e, the modes seem to go to the same input of the beam splitter network, and it is not clear what the cyan and red drawn on top of each other (in the very middle part of figure 1e) actually means. It looks like they are the same. If so, why should something different happen in the right-hand part of the figure?

2. Smaller matters:

* In the Fig. 1 caption, the authors write "The chip (yellow box) monolithically integrates..." and the detectors are drawn inside the yellow box, despite the fact that they are not on chip. Please fix.

* After equation 3, the authors have added "In our experiment, the probabilities for each measurement are calculated by normalizing two-fold coincidence counts over all outcomes between a control port and the d target ports." This still does not address reviewer 1's concerns that it is ambiguous what these coincidences describe. How about "... between a control port and any one of the d target ports"?

* On page 7 where the authors added the description about σ_x tensor σ_x , it might be good to add a phrase saying that the circuit allows any local qubit rotation, and specifically it can implement this σ_x tensor σ_x one by ..."

* The paragraph "Imperfect device fabrication..." in the discussion contains all of the relevant information in condensed form (so that's good), but it isn't all phrased as clearly as it might be. Some of the phrases don't convey enough specific detail to make them completely clear. For example, "Background photon noises may also be induced when the residual bright light propagates through the chip, though they are negligible owing to the measurement of coincidence" doesn't make it clear that the photon noise comes from 4WM of the pump. There's similar considerations for the other sentences.

Reviewer #2:

I have been asked to review the revised manuscript and comment on the response of the authors to both referees 1 and 2.

First let me commend the authors on addressing the comments in detail. The manuscript is almost ready for acceptance. I list the remaining issues to be addressed before acceptance. None of these is a fundamental science problem. They are about how things are described in the manuscript.

We are grateful to the referee again for his/her time and expertise for reviewing our revised manuscript. The comments and suggestions have been invaluable to clarify and further strengthen our manuscript. We have fully addressed the comments point by point as below.

1. One **significant element** of the text still needs fixing. Although the authors have made it much clearer that the choice of which-path or interference experiments is not actively switched on the timescale of the photon propagation, and have generally improved the discussion, I believe the manuscript still lacks a crystal-clear statement of **exactly** how this entanglement works out in practice.

Let me explain further. Let's consider just the extreme cases of the which-path or interference experiments, and ignore the in-between case. Obviously "which path" or "interference" is selected by getting one or other outcome in a measurement of the control qubit in the relevant basis. In turn, these outcomes project the target state into taking one mode or another early in the circuit (red arrow or cyan arrow, before d-BS1, in fig 1e). What is still not crystal clear is this: where do those two modes go? Do they go into two different physical copies of circuit, one with the d-Hadamard d-BS2 and one without? Do they go into two different modes of the fan-out? If so, how does this make a difference? Presumably they don't go into the same input mode of d-BS1 as this breaks unitarity. What I believe really needs to be explained in laborious detail is what actually happens specifically to each of these modes—where they go and why—so that the operation of the right-hand two thirds of the apparatus can be fully understood. At the moment, I feel that many readers will find this unclear. In the current version of fig.1e, the modes seem to go to the same input of the beam splitter network, and it is not clear what the cyan and red drawn on top of each other (in the very middle part of figure 1e) actually means. It looks like they are the same. If so, why should something different happen in the right-hand part of the figure?

We thank the referee for pointing out the importance of clarifying how the generalised controlled-Hadamard operation exactly works in our integrated device. We have now re-drawn and updated Fig1.e in the 3D format, which allows us to unambiguously illustrate the whole scheme. As we have stated in the manuscript (page 4) and the referee has correctly described, our implementation of state-process entanglement adopted two physically different layers of quantum circuits, one with the d-Hadamard d-BS2 (bottom red layer) and one without the d-BS2 (upper green layer). The two processes preserves coherent quantum superposition by erasing the information of which layer (which process) the target photon takes. Either multipath wave nature or multimode particle nature is revealed, is determined by the state of the control qubit in the delay-choice manner (no which-process information is known until the photons are measured). We believe that the revised Fig1.e now, together with the caption and main text, have provided a clear statement of how the state-process entanglement scheme works.

2. Smaller matters:

* In the Fig. 1 caption, the authors write “The chip (yellow box) monolithically integrates...” and the detectors are drawn inside the yellow box, despite the fact that they are not on chip. Please fix.

We thank the referee for pointing out the error in the manuscript. Indeed, all SNSPDs are not integrated on the chip yet. In the updated Fig.1e, the yellowed box has been removed. The description of off-chip SNSPDs has been added in Fig.1e and its caption.

* After equation 3, the authors have added “In our experiment, the probabilities for each measurement are calculated by normalizing two-fold coincidence counts over all outcomes between a control port and the d target ports.” This still does not address reviewer 1’s concerns that it is ambiguous what these coincidences describe. How about “... between a control port and any one of the d target ports”?

We thank the referee for suggestions. In the revised manuscript, we have corrected this statement as: In experiment, we measured two-photon coincidences between a control port and any one of the d target ports. The probabilities were then calculated by normalizing over the d target ports.

* On page 7 where the authors added the description about σ_x tensor σ_x , it might be good to add a phrase saying that the circuit allows any local qubit rotation, and specifically it can implement this σ_x tensor σ_x one by ...”

We thank the referee for suggestions. In the revised manuscript, we have emphasized this point as: “Our circuit allows any local qubit rotation by the MZIs together with posterior phase-shifters, see Fig.1e. In particular, local Pauli $\sigma_x \otimes \sigma_x$ operations on the Bell state were implemented to perform measurements in the $\{|+\rangle, |-\rangle\}$ basis.”

* The paragraph “Imperfect device fabrication...” in the discussion contains all of the relevant information in condensed form (so that’s good), but it isn’t all phrased as clearly as it might be. Some of the phrases don’t convey enough specific detail to make them completely clear. For example, “Background photon noises may also be induced when the residual bright light propagates through the chip, though they are negligible owing to the measurement of coincidence” doesn’t make it clear that the photon noise comes from 4WM of the pump. There’s similar considerations for the other sentences.

As the referee suggested, we have added more details in the discussion section:

“...For example, photon leakages between neighboring paths in the d-MZI, owing to the presence of thermal crosstalks and imperfect extinction ratio of 2-MZIs (~ 30 dB), brings in noises in the measurement of high-order interference. Further optimisations of device fabrication and alleviations of thermal crosstalk can improve the performance. Moreover, accidental counts may be induced from the undesirable SFWM process when the residual bright light propagates through the while chip, i.e., the parts after the sources, though they are negligible in the measurement of coincidences...”

We believe the referee’s concerns have been all addressed here and also in the revised manuscript (red colored text). We again thank the reviewer for his/her constructive comments and for the opportunity to further improve the presentation of our manuscript.

Reviewers' Comments:

Reviewer #2:

Remarks to the Author:

The authors have addressed all of the remaining comments and the manuscript is now suitable for publication (except for a few typos which will no doubt be addressed in proofs).

I thank the authors for the new Fig. 1e and revised caption. It addresses my corresponding concern completely, and makes the physical process very clear.